# The Moral, Ethical, Personal, and Professional Challenges Faced by Physicians during the COVID-19 Pandemic

**DOI:** 10.3390/ijerph19095641

**Published:** 2022-05-05

**Authors:** Krzysztof Goniewicz, Mariusz Goniewicz, Anna Włoszczak-Szubzda, Dorota Lasota, Frederick M. Burkle, Marta Borowska-Stefańska, Szymon Wiśniewski, Amir Khorram-Manesh

**Affiliations:** 1Department of Security, Polish Air Force University, 08-521 Dęblin, Poland; 2Department of Emergency Medicine, Medical University of Lublin, 20-093 Lublin, Poland; mariusz.goniewicz@umlub.pl; 3Faculty of Human Sciences, University of Economics and Innovation, 20-209 Lublin, Poland; anna.wloszczak-szubzda@wsei.lublin.pl; 4Department of Experimental and Clinical Pharmacology, Medical University of Warsaw, 02-097 Warsaw, Poland; dlasota@wum.edu.pl; 5Harvard Humanitarian Initiative, T.H. Chan School of Public Health, Harvard University, Boston, MA 02115, USA; skipmd77@aol.com; 6Institute of the Built Environment and Spatial Policy, University of Lodz, 90-142 Łódź, Poland; marta.borowska@geo.uni.lodz.pl (M.B.-S.); szymon.wisniewski@geo.uni.lodz.pl (S.W.); 7Department of Surgery, Institute of Clinical Sciences, Sahlgrenska Academy, Gothenburg University, 41345 Gothenburg, Sweden; amir.khorram-manesh@surgery.gu.se; 8Learning and Leadership for Healthcare Professional, Institute of Health and Care Sciences, Sahlgrenska Academy, Gothenburg University, 40530 Gothenburg, Sweden; 9Gothenburg Emergency Medicine Research Group (GEMREG), Sahlgrenska Academy, Gothenburg University, 40530 Gothenburg, Sweden

**Keywords:** COVID-19, physician, professional attitude, ethics

## Abstract

The COVID-19 pandemic, apart from the main problems concerning the health and life of patients, sparked a discussion about physicians’ moral and social professional attitudes. During a pandemic, physicians have the same ethical, moral, and medical responsibilities, however, the situation is different since they are self-exposed to a danger, which may influence their willingness to work. The problem of the professional moral attitudes of health care workers, recurring in ethical discussions, prompts us to define the limits of the duties of physicians in the event of a pandemic, hence this research aimed to assess these duties from an ethical perspective and to define their boundaries and scope. The study was conducted in May and June 2020 in the city of Lublin, covering all medical centers, and the questionnaire was completed by 549 physicians. The research was conducted in four areas: emergency standby in the event of a disaster, even if it is not requested; willingness to work overtime in the event of a disaster, even without payment; willingness to take health risks by caring for people who are infectious or exposed to hazardous substances; readiness to be transferred to other departments in the event of a disaster. Although most of the respondents declared to be agreed on personal sacrifices in the performance of professional duties, they were not prepared for a high level of personal risk when working in a pandemic. Excessive workload, its overwhelming nature, and personal risk are not conducive to readiness to work overtime, especially without pay. Research shows how important it is to respect the rights and interests of all parties involved in a pandemic. Physicians’ duty to care for a patient is also conditioned by the duty to protect themselves and should not be a tool for intimidating and depersonalizing their social and professional lives.

## 1. Introduction

While immediately perilous to the health and lives of infected patients, the SARS-CoV-2 (COVID-19) pandemic also raised both crucial and yet unknown dangers that could impede the safety of caretakers, leaving them feeling uncertain, unsafe, and vulnerable. These resulted in a myriad of critical discussions among providers regarding moral and professional obligations for care [1], raising several important questions. When can physicians refuse to perform professional duties? Where is the line between a physician’s right to refuse unsafe work and their duty to care for their patients? Can the health and life of your own and the physician’s family be a reason for not providing medical services? Could inadequate pay motivate why physicians leave their job during a pandemic? Does working in a pandemic have to mean 100% availability, including changing jobs or positions? [2,3,4,5].

The main ethical principles [6] which are to guide the medical community in treating patients are: Autonomy, i.e., the patient’s right to make medical decisions; Charity, i.e., relieving suffering, reducing pain, obtaining beneficial results, and improving the patient’s quality of life by medical personnel; No Harm, a physician should always act in “good faith”; Compassion, i.e., developing empathy and treating the patient with respect; and Justice, that is, fair distribution of funds available to a physician in their health care system among their patients [7].

The areas of the professional duty of physicians include four levels, i.e., moral duty, duty as a response to given social trust, duty as a professional norm defined by the labor code, and duty as a behavioral norm (socially sanctioned). Additionally, the social role of the “physician” is associated with sacrifice, that is, the duty of persistent fulfilment of one’s role and diligence, i.e., performing the duty with perseverance, regardless of the reward [7,8].

In the era of a pandemic, the duties of physicians are most often based on the so-called “special” duties or “physician role” duties. This is justified by special skills acquired as part of medical education, a voluntary decision to enter the medical profession with the awareness of what the job involves and the risks it entails, and the social contract between health professionals and society.

Several types of professional attitudes of physicians in the event of a pandemic have been distinguished, mainly:A “hero” physician, standing on the front line of the pandemic, regardless of the circumstances surrounding his work.A “professional” physician, seeing limitations in his work, e.g., in the absence of appropriate personal protective equipment and the quality of patient care.A physician who refuses to treat infected people.A physician quitting the profession in a pandemic [9].

It seems as though physicians do not have any rights, especially in a pandemic. However, most of the ethical codes emphasize that they do have rights, apart from the main obligations to the patients, society, and colleagues [10]. From this perspective, the obligation to treat cannot be “absolute”, stating that physicians are obliged to work under any circumstances. Physicians have the right to protection and care during an outbreak of infectious disease, as do other members of society [11]. Provision of care is possible under the assumption that physicians are healthy themselves. For a virus as dangerous as SARS-CoV-2, and when health systems have limited resources, overwhelmed physicians can simply be exposed to dangers during exercising their responsibilities. The problem of the professional moral attitudes of health care workers, recurring in ethical discussions, prompts us to define the limits of a physician’s duties in the event of a pandemic.

The current COVID-19 pandemic may have impacted the working situation for all healthcare professionals in various ways. In Ireland, it was a blessing since it improved the working conditions, as described by interviewees in a study that recognized more doctors staffing the hospital wards during the first wave of the pandemic, and other positive implications for a range of factors crucial for their experience of work, such as the ability to take sick leave, workplace relationships, collective workplace morale, access to senior clinical support, and the speed of clinical decision making [12]. These organizational improvements have been reported by other researchers, who also mentioned several changes in the physicians’ psychosocial work environment due to an increased workload and information overload, as well as ethical considerations and uncertainty, making the working environment stressful for physicians [13]. Additionally, a recent publication investigating the ethical viewpoints of both civilian and military populations facing diverse scenarios, including pandemics, showed differences between both populations, but also within each group. One significant factor influencing ethical viewpoints was the participants’ nationality (Polish vs. Swedish), but also gender and professions [14].

Therefore, this work aims to examine the professional attitudes of physicians based on age and seniority at the time of the COVID-19 pandemic in terms of moral, ethical, professional, and social standards in Lublin, Poland.

## 2. Materials and Methods

### 2.1. Location of the Study

The study was conducted in May and June 2020 in the city of Lublin and included all medical centers, i.e., hospitals, institutes, and the Medical University.

### 2.2. Questionnaire

The cross-sectional survey was developed based on a literature review by all authors. The following search engines were used to search for literature: PubMed, Scopus, and Web of Science. Using the following keywords: “COVID-19” AND “ethics” AND “professional attitude” AND “physician”, a high number of hits was obtained. The search was then limited to the publications in English, then researchers conducted the search independently, and the outcomes were then matched.

Included studies: Original publications and reviews from January 2010 to 2022.

Excluded studies: Proceedings, editorials, meeting notes, news, abstracts, and non-relevant papers.

Finally, a qualitative thematic analysis of the included literature based on an inductive approach was applied. This content analysis aimed to study all included articles, focusing on similarities and differences in the findings to build the questionnaire.

The acquired data were then organized, categorized, and mapped. The questionnaire consisted of 9 questions and was constructed to be completed in 5 min. There were four questions, which aimed to assess the perceived preparedness quantitatively. These questions aimed to quantify perceived ethical issues. Each question in this group was formulated as a statement which could be answered using a Likert scale from 1 to 5, where 5 meant ‘agree’, 4 ‘disagree’, and 5 ‘no opinion’. The remaining questions were demographic, and the following variables were collected: age, gender, workplace, and length of service.

To verify the research tool, the questionnaire was tested on a sample of 15 employees in one university hospital. This group was then excluded from the study and their answers were not included in the final analysis. The outcome was reviewed based on a combination of logic, relevance, comprehension, legibility, clarity, and usability. In the initial phase of the tool verification, the questionnaire was tested on 20 physicians of the Medical University of Lublin who were then excluded from the main study.

### 2.3. Data Collection and Study of Population

Due to the prevailing COVID-19 pandemic, the survey was available online. The online version was sent to physicians employed at the Medical University of Lublin and the relevant authorities were asked for permission. The questionnaire was presented in a google format. In total, five hundred and forty-nine physicians completed the questionnaire.

### 2.4. Statistical Analysis

The statistical analyses were carried out with IBM SPSS Statistics version 23. It was used for frequency analysis and basic descriptive statistics: multi-field contingency tables, graphs, and hypothesis testing. For qualitative features, the Chi^2^ test was used to detect the existence of a relationship between the analyzed variables. Due to the small numbers in the tables, the Yates continuity correction was applied. The classical statistical significance level was adopted as *p* ≤ 0.05.

### 2.5. Ethical Considerations

The information included the study’s purpose, the voluntary nature of their participation, and strict confidentiality and secure data storage. The survey had anonymous nature and all respondents agreed to participate in the survey. The study is not a medical experiment and legally does not require the opinion of the Bioethics Committee, according to Polish Law.

## 3. Results

Of all the physicians, 54.3% were men. Similar values were recorded for all age ranges. Of the respondents, 79.8% worked in a public hospital, and 35.5% of the physicians had been working for more than 20 years, while 14.8% had up to 5 years of service. The results are presented in Table 1.

The review resulted in accumulated data that could be sorted, analyzed, and then categorized into four diverse areas: readiness to be on duty in the event of a disaster, at the workplace, even if it was not requested; willingness to work overtime in the event of a disaster, even without payment; willingness to take health risks by caring for people who are infectious or exposed to hazardous substances; and readiness to be transferred to other departments in the event of a disaster.

In the case of a statement regarding the readiness to report on-call duty in the event of a disaster, even if the respondents were not asked to do so, 64% of them agreed with such an attitude.

With the statement regarding the readiness to work overtime without payment in the event of a disaster, 54% of the physicians did not agree to it.

If they are ready to take health risks, when caring for people suffering from infectious diseases or exposed to dangerous substances, the respondents most often agreed with 45%, N = 248. Subsequently, the respondents were to refer to the statement regarding the readiness to be transferred to other departments in the event of a disaster. The respondents most often chose the answer to a comparable agree (37%, N = 201) and disagreed (39%, N = 214) (Figure 1).

In all four research areas, only the dependencies on the variables of the age of the respondents and their length of service turned out to be statistically highly significant (*p* ≤ 0.000).

When it comes to the age of the respondents and their declaration of reporting on duty in the event of a disaster, the respondents aged 55 and more most often agreed (N = 104, 74%, rather agree/agree). Most often, the respondents aged up to 34 did not have an opinion on this subject (N = 55, 43%).

Most often, respondents from the 45–54 age group (N = 85, 63%) did not agree and rather did not agree to take up overtime work without payment in the event of a disaster. Most often, the respondents aged up to 34 did not have an opinion about taking up overtime work in the event of a disaster (even without payment) (N = 54, 43%).

The oldest respondents (55 and over) most often declared readiness to take health risks when caring for people suffering from infectious diseases or exposed to dangerous substances (N = 78, 55%, rather agree/agree). The answer “I have no opinion” was most often chosen by the youngest respondents—up to 34 years of age (N = 37, 30%).

The youngest respondents (up to 34 years of age), when asked about their readiness to be transferred to other departments when emergency assistance is needed, most often answered: “I have no opinion” (N = 46, 36%). Most often, the respondents aged 35–44 years old did not agree to such a transfer (N = 64, 43%, rather disagree/disagree) (Table 2).

The respondents whose work experience was in the range of 16–20 years agreed more often than the rest of the respondents to take up on-call duty in the event of a disaster (N = 74, 74%, rather agree/agree). Most often, the respondents in the 6–10 years period of service (N = 18, 20%, I rather disagree/disagree) disagreed with this on-call duty. The respondents with the shortest seniority from 0–5 years most often declared no opinion on this issue (N = 36, 44%).

Much more, respondents with 0 to 5 years of service did not have an opinion on overtime work in the event of a disaster, even without payment (N = 36, 44%), and 6 to 10 years of service (N = 36, 40%). As a percentage, more than half of the respondents from two age groups—11–15 years (N = 49, 58%) and more than 20 years (N = 116, 59%)—did not agree to such work or rather did not agree.

Respondents with 11–15 years of work experience, much more often than the rest of the respondents, declared readiness to take health risks when caring for people suffering from infectious diseases or exposed to dangerous substances (N = 45, 53%, rather agree/agree). “I have no opinion” was most often answered by respondents with 6–10 years of experience (N = 33, 37%).

As for the work experience of the respondents and readiness to be transferred to other departments, when assistance in the event of a disaster is needed, most often, respondents with 11–15 years of experience did not agree (N = 42, 50%, I rather disagree/disagree). “I have no opinion” was most often answered by people with 6–10 years of work experience (N = 38, 43%), (Table 3).

The dependencies on the “gender” variable were statistically significant in the following areas:-“Willingness to work overtime in the event of a disaster, even without payment”, where most often men disagreed with overtime work, including without payment (N = 180, 60%, rather disagree/disagree). The relationship between gender and willingness to work overtime, even without payment, was statistically highly significant (*p* ≤ 0.001).-“Readiness to take health risks when caring for people with infectious diseases or exposed to dangerous substances”, where readiness to take health risks when caring for people with infectious diseases or exposed to dangerous substances was declared much more often by women (N = 125, 50%, rather I agree/I agree). Both genders answered the question “I have no opinion” (women N = 59, 24%; men N = 66, 22%). The relationship between gender and the readiness to take health risks while caring for people suffering from infectious diseases or exposed to dangerous substances was statistically significant (*p* ≤ 0.038).-The dependence on the variable “workplace” was statistically significant in only one area: “Readiness to take health risks when caring for people suffering from infectious diseases or exposed to hazardous substances”, where respondents employed in research units expressed their consent much more often (N = 70, 64%, I tend to agree/agree). The relationship between the respondents’ workplace and their willingness to take health risks while caring for people suffering from infectious diseases or exposed to dangerous substances was statistically highly significant (*p* ≤ 0.000).

## 4. Discussion

This study confirms the acceptance of personal sacrifice among Polish physicians during their professional duties. However, not everyone in this group was prepared to work unconditionally and accept the personal risk of infection during a pandemic. Similar results have been published for other healthcare professionals and necessitate new considerations regarding the professional working environment before an increasing number of future public health emergencies [15].

Pursuing a medical profession is associated with social trust as well as a duty and/or sacrifice. Therefore, it is commonly believed that the ideological foundation of the medical profession lies in the “altruism”, which is defined as giving priority to the benefits of others, with the possibility of one’s losses [16,17]. This belief is also recognized in the Hippocratic Oath, which is the basis of today’s medical ethics, requiring a physician to commit: “I will use ... my greatest abilities and judgment, and I will not do any harm or injustice” [18]. The American College of Physicians Handbook of Ethics states further that the ethical imperative of providing physician care overrides the risks to the physician even in a pandemic. The American Society of Infectious Diseases, as well as the American Medical Association (AMA), also emphasize physicians’ obligation to treat, even at the risk of contracting the patient’s disease, or when faced with greater than usual risks to their safety, health, or life [19,20]. Similar guidelines can be found in Good Medical Practice in Great Britain, where the General Medical Council obliges the physician to treat cases of high risk [21].

Although there are several guidelines and instructions, the perception of those facing the real threat at the operational level should also be considered. In a study by Sultan et al. [22], there were clear differences in healthcare staff’s willingness to work in different disaster and public health emergency scenarios. Additionally, Khorram-Manesh et al. have shown that ethical and moral viewpoints of both civilian and military healthcare staff facing various scenarios are not predictable, emphasizing a need for both synchronizing these views between diverse agencies and in each population [23]. In a study by Shabanowitz et al., most employees (60%) believed that giving up their jobs during a pandemic is unethical because of their duty to care, while 65% wanted autonomy in deciding whether to work or not, although 79% would agree to volunteer with some incentives such as protective equipment and training in communicable diseases [24]. In the same study, 64% of respondents declared their readiness to report on-call duty in the event of a disaster, even if the respondents were not asked to do so. Most often they were respondents aged 55 and more (74%). The respondents whose length of service was in the range of 16–20 years (74%) also agreed more often than the rest of the respondents to take on duty in the event of a disaster [21,24].

Planning work in a pandemic situation is a real challenge for management. An excessive workload, its overwhelming nature (moral dilemmas, stress, fear and anxiety, insomnia), and personal risk (fear of self-contamination) do not favor readiness to work overtime, especially without remuneration [21]. Research by Alanezi Fahad and co-authors shows that the main challenge for physicians in the pandemic in Saudi Arabia was an excessive workload. Therefore, most of the respondents would prefer to reduce their working hours and minimize the number of night shifts [25]. When looking at the readiness to work overtime without payment in the event of a disaster, more than half of the physicians (54%) did not impress. They were most often respondents from the 45–54 age group (63%). More than half of the respondents in the length of service groups 11–15 (58%) and >20 years (59%) were of a similar opinion. Such readiness was most often refused by men (60%).

Undoubtedly, the COVID-19 pandemic posed a great challenge to the altruistic attitude of medical personnel towards patients, making the autonomy of medical personnel in the context of patient care obligations a controversial issue [26,27]. The COVID-19 pandemic has indeed illustrated that healthcare professionals are not and cannot be obligated to do absolutely everything in their power to benefit their patients at any level of personal risk. For instance, employees suffering from conditions that increase their risk of COVID-19 were advised to avoid contact with patients, an unacceptable level of personal risk, and unnecessary heroism [28,29].

Generally, in modern healthcare, the risk of exposure to infectious diseases is not ubiquitous, so a physician may argue that occupational exposure to pathogens is not an integral part of his or her normal work [29,30,31]. Empirical data on healthcare professionals’ attitudes to personal risk and responsibilities show that not every worker feels comfortable accepting such risks. For instance, in an American study, only 55% of physicians agreed that a physician must care for a patient during an outbreak, even if it endangers the physician’s health. Furthermore, a British study found that 26% of health care workers disagreed that all health professionals have a duty to work, even if the working environment is exposed to high risk. Therefore, it cannot be assumed that all health care professionals are prepared for a high level of personal risk when working in a pandemic [32,33]. This statement is confirmed by the results of the study conducted by Ayub et al. on the attitudes of physicians toward treating COVID-19 patients in Pakistan, when as many as 83% of 208 participants expressed their reluctance to treat patients with COVID-19, arguing their position with fear of self-infection, and infecting family members [34]. At the same time, 45% of the respondents, mostly the oldest respondents, in age group of 55 and more (55%), expressed their readiness to care for people suffering from infectious diseases or exposed to hazardous substances. Such readiness was much more often declared by respondents with 11–15 years of work experience (53%), women (50%), and respondents employed in research units (64%). The same results were presented by Sultan et al. [22], who also concluded that experience and theoretical knowledge increased the confidence of healthcare staff and their willingness to work under severe conditions.

The current COVID-19 pandemic enforced a rotation between diverse specialties to cover the need for physicians [35]. It also highlighted the moral and ethical issues of the treatment of patients infected with COVID-19 by physicians who lack the necessary knowledge and practical skills. As Dawid Orentlicher notes, not every physician can be an emergency room physician dealing with emergencies [36]. The lack of appropriate training is a significant challenge, especially for younger physicians, as indicated by Ayub et al. [34]. Therefore, physicians without specialist training and those who are uncertain about their competency should not be forced to participate in or be delegated duties related to the intensive care of a patient. Additionally, the World Health Organization (WHO) interim guidelines for COVID-19 allow health care workers to exercise their right to remove themselves from their work situation if they have reasonable grounds to do so [37]. The guideline is based on the opinions of healthcare workers on the readiness to be transferred to other departments in the event of a catastrophe. In this study, 37% of the respondents agreed, and 39% did not agree to be transferred. Most often, respondents aged 35–44 (43%) and respondents with 11–15 years of experience (50%) did not consent to such a transfer.

In summary, although healthcare professionals have an obligation to get involved in the treatment of patients in hazardous environments and severe conditions, such as those with COVID-19, the capabilities and limitations of such involvement should be critically assessed, making such involvement conditional. Research shows how important it is to respect the rights and interests of all parties involved in a hazardous situation such as a pandemic. The core principle in working disasters and emergencies is the Major Incident Medical Management Support (MIMMS) three S’s in safety: Self Safety, Scene Safety, and Survivors Safety [38]. An unsafe healthcare provider cannot be of any help to an incident survivor. Therefore, a physician’s duty to care for the patient is also conditioned by the duty of self-protection. This should be considered as their right to refuse to work in unfavorable conditions unless appropriate safety and security are provided. Physicians are required to be involved in a pandemic response because of their specific abilities, but these abilities vary from physician to physician [39,40,41,42,43,44,45]. In circumstances such as a pandemic, some responsibilities may be considered “supererogation” or good, but not morally required. From an ethical and pragmatic point of view, physicians should be viewed in the context of their entire life, including their personal and professional lives.

## 5. Limitations

The main limitation of this study was the limited number of physicians from one city (the city of Lublin). The COVID-19 pandemic proved to be an obstacle to further research, which could have impacted response rates and might have generated response bias. Another limitation was the lack of consideration for illness, personal or chronic conditions, pregnancy, pressure from family or spouse, or other factors that might have influenced the responses. Additionally, there was no space for free text in the questionnaire, thus respondents did not have the opportunity to add topics they felt were of relevance. Another weakness of the study was determining the participation rate and assessing the representativeness of the sample as the number of physicians eligible for the study has not been specified. Despite these limitations, this study opens a discussion on this subject and the need for broader research in this area. Due to the essence of the problem, and its related consequences, further in-depth research is recommended. At the same time, this study serves as a wider standardization of the research tool used.

## 6. Conclusions

In conclusion, most of the respondents, in this study, agreed to personal sacrifice when conducting their professional duties. However, not all healthcare professionals are prepared for a high level of personal risk when working in a pandemic. An excessive workload, its overwhelming nature, and personal risk are not conducive to readiness to work overtime, especially without pay. It is the responsibility of healthcare managers to provide protective, psychological, and other necessary support to physicians and other healthcare staff involved in responding to the pandemic and other public health emergencies.

Improving human resource management in health care during a pandemic should include monitoring the “healthy worker effect”, improving interpersonal relationships (e.g., through support groups), and supporting the “command-and-distribution” system with an individual approach to the physician. The ethics of health workers, safety, and autonomy in the care of patients in public health emergencies, require further research.

## Figures and Tables

**Figure 1 ijerph-19-05641-f001:**
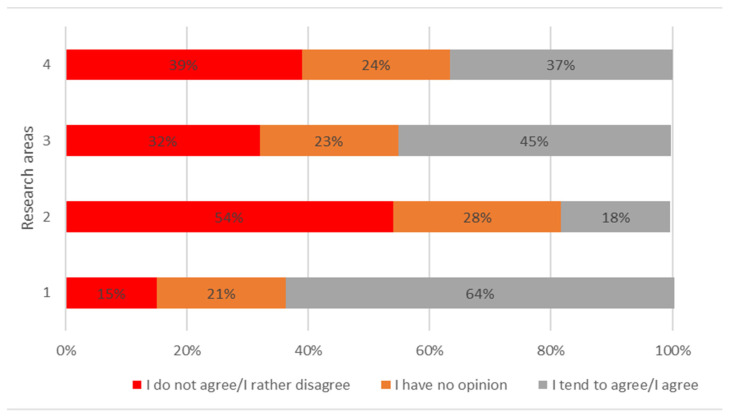
Percentage distribution of respondents’ answers in individual research areas 1–4. (1) Readiness to report to work in the event of a workplace disaster; (2) Willingness to work overtime in the event of a disaster, even without payment; (3) Willingness to take health risks when caring for people who are infectious or exposed to hazardous substances; (4) Willingness to transfer to other departments when disaster relief is needed.

**Table 1 ijerph-19-05641-t001:** Sociodemographic data.

**Gender**	**N**	**%**
Female	251	45.7%
Male	298	54.3%
Total	549	100%
**Age**	**N**	**%**
Up to 34	127	23.1%
35–44 years	147	26.8%
45–54 years	134	24.4%
55 years and over	141	25.7%
Total	549	100%
**Workplace**	**N**	**%**
Public hospital	438	79.8%
Research facility	110	20.0%
Other	1	0.2%
Total	549	100%
**Length of Service**	**N**	**%**
From 0 to 5 years	81	14.8%
6–10 years	89	16.2%
11–15 years	84	15.3%
16–20 years	100	18.2%
More than 20 years	195	35.5%
Total	549	100%

**Table 2 ijerph-19-05641-t002:** Research areas and the age of respondents.

**Research Area/Respondents’ Answers**	**I. Readiness to Report to Work in the Event of a Workplace Disaster**									
	**Age**									
	**Up to 34**		**35–44 Years**		**45–54 Years**		**55 Years and Over**		**Total**	
	**N**	**%**	**N**	**%**	**N**	**%**	**N**	**%**	**N**	**%**
I have no opinion	55	43	22	15	19	14	21	15	117	21
I disagree	4	3	0	0	8	6	7	5	19	3
Rather disagree	13	10	27	18	14	11	9	6	63	12
Rather agree	43	34	88	60	75	56	70	50	276	50
I agree	12	10	10	7	18	13	34	24	74	14
Total	127	100	147	100	134	100	141	100	549	100
**Chi-Squared Test**										
**Value**			**df**				** *p* **			
**Chi^2^**	**Yates’ Correction for Continuity**		**Chi^2^**	**Yates’ Correction for Continuity**			**Chi^2^**	**Yates’ Correction for Continuity**		
84.173	76.864		12	12			0.000	0.000		
**Research Area/Respondents’ Answers**	**II. Willingness to Work Overtime in the Event of a Disaster, Even without Payment**									
	**Up to 34**		**35–44 Years**		**45–54 Years**		**55 Years and Over**		**Total**	
	**N**	**%**	**N**	**%**	**N**	**%**	**N**	**%**	**N**	**%**
I have no opinion	54	43	42	28	27	20	29	21	152	28
I disagree	14	11	30	20	31	23	26	18	101	18
Rather disagree	42	33	45	31	54	40	54	38	195	36
Rather agree	4	3	26	18	14	11	23	16	67	12
I agree	13	10	4	3	8	6	9	7	34	6
Total	127	100	147	100	134	100	141	100	549	100
**Chi-Squared Test**										
**Value**			**df**				** *p* **			
**Chi^2^**	**Yates’ Correction for Continuity**		**Chi^2^**	**Yates’ Correction for Continuity**			**Chi^2^**	**Yates’ Correction for Continuity**		
44.228	39.254		12	12			0.000	0.000		
**Research Area/Respondents’ Answers**	**III. Willingness to Take Health Risks When Caring for People Who Are Infectious or Exposed to Hazardous Substances**									
	**Up to 34**		**35–44 Years**		**45–54 Years**		**55 Years and Over**		**Total**	
	**N**	**%**	**N**	**%**	**N**	**%**	**N**	**%**	**N**	**%**
I have no opinion	37	30	36	24	26	19	26	18	125	23
I disagree	4	3	10	7	24	18	15	11	53	10
Rather disagree	41	32	34	23	26	19	22	16	123	22
Rather agree	41	32	60	41	49	37	58	41	208	38
I agree	4	3	7	5	9	7	20	14	40	7
Total	127	100	147	100	134	100	141	100	549	100
**Chi-Squared Test**										
**Value**			**df**				** *p* **			
**Chi^2^**	**Yates’ Correction** **for Continuity**		**Chi^2^**	**Yates’ Correction for Continuity**			**Chi^2^**	**Yates’ Correction for Continuity**		
45.078	39.524		12	12			0.000	0.000		
**Research Area/Respondents’ Answers**	**IV. Willingness to Transfer to Other Departments When Disaster Relief Is Needed**									
	**Up to 34**		**35–44 Years**		**45–54 Years**		**55 Years and Over**		**Total**	
	**N**	**%**	**N**	**%**	**N**	**%**	**N**	**%**	**N**	**%**
I have no opinion	46	36	32	22	26	19	30	22	134	24
I disagree	8	6	24	16	26	19	20	14	78	14
Rather disagree	37	29	40	27	26	19	33	23	136	25
Rather agree	33	26	48	33	51	39	41	29	173	32
I agree	3	3	3	2	5	4	17	12	28	5
Total	127	100	147	100	134	100	141	100	549	100
**Chi-Squared Test**										
**Value**			**df**				** *p* **			
**Chi^2^**	**Yates’** **Correction** **for Continuity**		**Chi ^2^**	**Yates’ Correction for Continuity**			**Chi^2^**	**Yates’ Correction for Continuity**		
43.032	37.499		12	12			0.000	0.000		

Note: Due to small numbers in some cells in the table, the Yates continuity correction was applied.

**Table 3 ijerph-19-05641-t003:** Research areas and seniority.

**Research Area/Respondents’ Answers**	**I. Readiness to Report to Work in the Event of a Workplace Disaster**											
	**Length of Service**											
	**From 0 to 5 Years**		**6–10 Years**		**11–15 Years**		**16–20 Years**		**More than 20 Years**		**Total**	
	**N**	**%**	**N**	**%**	**N**	**%**	**N**	**%**	**N**	**%**	**N**	**%**
I have no opinion	36	44	26	29	12	14	14	14	29	15	117	21
I disagree	0	0	4	4	0	0	4	4	11	6	19	3
Rather disagree	11	14	14	16	15	18	8	8	15	8	63	11
Rather agree	29	36	38	43	47	56	68	68	94	48	276	50
I agree	5	6	7	8	10	12	6	6	46	23	74	13
Total	81	100	89	100	84	100	100	100	195	100	549	100
**Chi-Squared Test**												
**Value**			**df**				** *p* **					
**Chi^2^**	**Yates’ Correction for Continuity**		**Chi^2^**	**Yates’ Correction for Continuity**			**Chi^2^**			**Yates’ Correction for Continuity**		
83.853	73.923		16	16			0.000			0.000		
**Research Area/Respondents’ Answers**	**II. Willingness to Work Overtime in the Event of a Disaster, Even without Payment**											
	**From 0 to 5 Years**		**6–10 Years**		**11–15 Years**		**16–20 Years**		**More than 20 Years**		**Total**	
	**N**	**%**	**N**	**%**	**N**	**%**	**N**	**%**	**N**	**%**	**N**	**%**
I have no opinion	36	44	36	40	16	19	26	26	38	20	152	28
I disagree	9	11	5	6	27	32	30	30	30	15	101	18
Rather disagree	28	35	36	40	22	26	23	23	86	44	195	36
Rather agree	4	5	3	4	15	18	17	17	28	14	67	12
I agree	4	5	9	10	4	5	4	4	13	7	34	6
Total	81	100	89	100	84	100	100	100	195	100	549	100
**Chi-Squared Test**												
**Value**			**df**						** *p* **			
**Chi^2^**	**Yates’ Correction for Continuity**		**Chi^2^**		**Yates’ Correction for Continuity**				**Chi^2^**		**Yates’ Correction for Continuity**	
76.402	67.622		16		16				0.000		0.000	
**Research Area/Respondents’ Answers**	**III. Willingness to Take Health Risks When Caring for People Who Are Infectious or Exposed to Hazardous Substances**											
	**From 0 to 5 Years**		**6–10 Years**		**11–15 Years**		**16–20 Years**		**More than 20 Years**		**Total**	
	**N**	**%**	**N**	**%**	**N**	**%**	**N**	**%**	**N**	**%**	**N**	**%**
I have no opinion	23	28	33	38	13	16	18	18	38	19	125	23
I disagree	4	5	0	0	10	12	20	20	19	10	53	10
Rather agree	22	27	34	38	16	19	20	20	31	16	123	22
Rather disagree	32	40	18	20	38	45	42	42	78	40	208	38
Rather agree	0	0	4	4	7	8	0	0	29	15	40	7
I agree	81	100	89	100	84	100	100	100	195	100	549	100
**Chi-Squared Test**												
**Value**			**df**						** *p* **			
**Chi^2^**	**Yates’ Correction for Continuity**		**Chi^2^**		**Yates’ Correction for Continuity**				**Chi^2^**		**Yates’ Correction for Continuity**	
89.025	79.223		16		16				0.000		0.000	
**Research Area/Respondents’ Answers**	**IV. Willingness to Transfer to Other Departments When Disaster Relief Is Needed**											
	**From 0 to 5 Years**		**6–10 Years**		**11–15 Years**		**16–20 Years**		**More than 20 Years**		**Total**	
	**N**	**%**	**N**	**%**	**N**	**%**	**N**	**%**	**N**	**%**	**N**	**%**
I have no opinion	25	31	38	43	13	16	13	13	45	23	134	24
I disagree	4	5	4	5	22	26	22	22	26	13	78	14
Rather disagree	22	27	32	36	20	24	17	17	45	23	136	25
Rather agree	30	37	12	13	27	32	45	45	59	31	173	32
I agree	0	0	3	3	2	2	3	3	20	10	28	5
Total	81	100	89	100	84	100	100	100	195	100	549	100
**Chi-Squared Test**												
**Value**			**df**						** *p* **			
**Chi^2^**	**Yates’ Correction for Continuity**		**Chi^2^**		**Yates’ Correction for Continuity**				**Chi^2^**		**Yates’ Correction for Continuity**	
85.598	75.888		16		16				0.000		0.000	

Note: Due to small numbers in some cells in the table, the Yates continuity correction was applied.

## Data Availability

Not applicable.

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
