# Peer review of "The Moral, Ethical, Personal, and Professional Challenges Faced by Physicians during the COVID-19 Pandemic"

_ijerph, 2022, doi:10.3390/ijerph19095641_

Round 1
Reviewer 1 Report
Dear authors,
Thank you very much for the opportunity to review this review. I read the work with great interest. It carries a lot of useful information, however, before making a decision, many inaccuracies should be clarified:
INTRODUCTION
- The introduction could be enriched with a paragraph dealing with the specific working conditions of physicians during a pandemic before presenting the problematic of the study.
- The authors should specify in the objectives they will evaluate whether there are differences in attitudes of physician based on age and seniority.
MATERIALS AND METHODS
l 95:
- The authors do not specify the type of study in the methodology.
- The authors could specify the key words used in the literature review before developing the questionnaire
- Information about the data collected in the questionnaire is missing (description of variables collected, number of questions)
l 105-108
- The authors do not specify whether the questionnaire is anonymous and whether participation in the study is voluntary
- Details of the software used for the online questionnaire are missing. The number of physicians eligible for the study should be specified to assess the participation rate. Is the sample representative of the source population ( about age, gender, workplace)?
l109-115
- The authors could have specified that they would investigate whether there was a significant difference attitudes of physicians by age group and by seniority.
RESULTS
Figure 1:
- The meaning of the research areas is missing
- l149: The value of the p-value is to be specified
- l155-166, Tables 2 and 3 : The authors could have grouped classes ( 3: I have no opinion; I disagree/Rather disagree; I rather agree/ I agree) to facilitate the interpretation of the results.
- l124-214: There are redundancies between the results presented in the tables and the text. The authors should review the presentation of the results
DISCUSSION:
- The authors could highlight the results of their study more for discussion
- The authors could enrich the discussion by presenting the differences observed in the study according to age and seniority.
- The authors could discuss the weaknesses of the study in relation to the choice of a cross-sectional survey, the representativeness of the sample, the declarative mode of information collection.
Author Response
Response to Reviewer #1
Manuscript ID: ijerph-1699808
Title: The Moral, Ethical, Personal, and Professional Challenges Experienced by Physicians during the COVID-19 Pandemic
Authors: Krzysztof Goniewicz, Mariusz Goniewicz, Anna Włoszczak-Szubzda, Dorota Lasota Frederick M. Burkle, Marta Borowska-Stefańska, Szymon Wiśniewski and Amir Khorram-Manesh
May 2015
Dear Editor,
We would like to express our sincere thanks to you for the positive and constructive comments on our previous manuscript. Based on your comments, we have worked to improve the paper. Our response is provided below, with reviewer comments in italics followed by our response in plain text.
Best regards,
Authors
Comment 1 INTRODUCTION: The introduction could be enriched with a paragraph dealing with the specific working conditions of physicians during a pandemic before presenting the problematic of the study. The authors should specify in the objectives they will evaluate whether there are differences in attitudes of physician based on age and seniority.
Response: Many thanks for your thoughts. We agree with this suggestion. In our revised paper, we have added information that, in our opinion, strengthens the discussion.
At the beginning we added “While immediately perilous to the health and lives of infected patients the SARS-COV-2 (COVID -19) pandemic also raised both crucial and yet unknown dangers that could impede the safety of caretakers leaving them feeling uncertain, unsafe, and vulnerable. These resulted in a myriad of critical discussions among providers regarding moral and professional obligations for care [1],”
And the discussion was also extended by a paragraph referring to
“…The current COVID-19 pandemic may have impacted the working situation for all healthcare professionals in various ways. In Ireland, it was a blessing since it improved the working condition as described by interviewees in a study that recognized more doctors staffing the hospital wards during the first wave of the pandemic, and other positive implications for a range of factors crucial for their experience of work, such as the ability to take sick leave, workplace relationships, collective workplace morale, access to senior clinical support and the speed of clinical decision-making [12]. These organizational improvements have been reported by other researchers, who also mentioned several changes in physicians ’psychosocial work environment due to increased workload, and information overload, as well as ethical considerations, uncertainty, making working environment stressful for physicians [13]. Additionally, a recent publication, investi-gating the ethical viewpoints of both civilian and military populations facing diverse scenarios, including pandemics, showed differences between both populations but also within each group. One significant factor influencing ethical viewpoints was participants' nationality (Polish vs. Swedish), but also gender, and professions [14]…’
Comment 2: MATERIALS AND METHODS
The authors do not specify the type of study in the methodology.
The authors could specify the key words used in the literature review before developing the questionnaire
Information about the data collected in the questionnaire is missing (description of variables collected, number of questions)
The authors do not specify whether the questionnaire is anonymous and whether participation in the study is voluntary
Details of the software used for the online questionnaire are missing. The number of physicians eligible for the study should be specified to assess the participation rate. Is the sample representative of the source population ( about age, gender, workplace)?
Response: Thanks for pointing this out. We agree to these comments. First the methodology was expanded, and we explained missing details. We have added the keys just like the excluded and included criteria. As for the size of the sample, it is something that is hard to explain. The questionnaire was sent by the internal system of the medical university in which the doctors have registered accounts. We are not able to determine how many such people are there due to the protection of personal data. It is also difficult to diagnose how many people actually received such a message - i.e. how many opened it or how many doctors simply ignored it.
We add this information into the text:
The questionnaire was developed based on a literature review by all authors. The following search engines were used to search for literature: PubMed, Scopus, and Web of Science. Using the following keywords: “COVID-19” AND “ethics” AND “professional attitude” AND “physician”, a high number of hits was obtained. The search was then limited to the publications in English, then researchers conducted the search inde-pendently, and the outcomes were then matched.
Included studies: Original publications and reviews from January 2010 to 2022
Excluded studies: Proceedings, editorials, meeting notes, news, abstracts, and non-relevant papers.
Finally, a qualitative thematic analysis of the included literature based on an in-ductive approach was applied. This content analysis aimed to study all included articles, focusing on similarities and differences in the findings to build the questionary.
The acquired data were then organized, categorized, and mapped. The question-naire-based on the descriptive research method consisted of 9 questions and was con-structed to be completed in 5 minutes. There were four questions, which aimed to assess the perceived preparedness quantitively. These questions aimed to quantify perceived ethical issues. Each question in this group was formulated as a statement, which could be answered using a Likert scale from 1 to 5, where 5 meant ‘agree’, 4 ‘disagree’ and 5 ‘no opinion’. The remaining questions were demographic. To verify the research tool, the questionnaire was tested on a sample of 15 employees in one university hospital. This group was then excluded from the study and their answers were not included in the final analysis. The outcome was reviewed based on a combination of logic, relevance, com-prehension, legibility, clarity, and usability. In the initial phase of the tool verification, the questionnaire was tested on 20 physicians of the Medical University of Lublin who were then excluded from the main study.
Comment 3: RESULTS
Figure 1:
The meaning of the research areas is missing
l149: The value of the p-value is to be specified
l155-166, Tables 2 and 3 : The authors could have grouped classes ( 3: I have no opinion; I disagree/Rather disagree; I rather agree/ I agree) to facilitate the interpretation of the results.
l124-214: There are redundancies between the results presented in the tables and the text. The authors should review the presentation of the results
Response: Thanks for these comments. We have revised results section add fixed all gaps. We also considered modifying the table but came to the conclusion that this form accurately portrays the differences in the results we wanted to present. Of course, we are aware that readability can be an obstacle, but on the other hand, our article is focused on a group of specialists and therefore it should not be a problem
Comment 4: DISCUSSION:
The authors could highlight the results of their study more for discussion
The authors could enrich the discussion by presenting the differences observed in the study according to age and seniority.
The authors could discuss the weaknesses of the study in relation to the choice of a cross-sectional survey, the representativeness of the sample, the declarative mode of information collection.
Response: Thank you very much for your suggestions. That was very important and we believe we have greatly strengthened the discussions with new paragraphs as well as modifications to the previous ones. The following parts has been added:
“…This study confirms the acceptance of personal sacrifice among Polish physicians during their professional duties. However, not everyone in this group was prepared to work unconditionally and accepted the personal risk of infection during a pandemic. Similar results have been published for other healthcare professionals and necessitate new considerations regarding the professional working environment before an increasing number of future public health emergencies [15].Pursuing a medical profession is associated with social trust as well as a duty and/or sacrifice. Therefore, it is commonly believed that the ideological foundation of the medical profession lies in the “altruism”, which is defined as giving priority to the benefits of others, with the possibility of one's losses [16-17]. This belief is also recognized in the Hippocratic Oath, which is the basis of today's medical ethics, requiring a physician to commit: "I will use ... my greatest abilities and judgment, and I will not do any harm or injustice" [18]. The American College of Physicians Handbook of Ethics states further that the ethical imperative of providing physician care overrides the risks to the physician even in a pandemic. The American Society of Infectious Diseases, as well as the American Medical Association (AMA), also emphasize physicians’ obligation to treat, even at the risk of contracting the patient's disease, or when faced with greater than usual risks to their safety, health, or life [19-20]. Similar guidelines can be found in Good Medical Practice in Great Britain, where the General Medical Council obliges the physician to treat cases of high risk [21].Although there are several guidelines and instructions, the perception of those facing the real threat at the operational level should also be considered. In a study by Sultan et al. [22], there were clear differences in healthcare staff’s willingness to work in different disaster and public health emergency scenarios. Additionally, Khorram-Manesh et al. have shown that ethical and moral viewpoints of both civilian and military healthcare staff facing various scenarios are not predictable, emphasizing a need for both synchronizing these views between diverse agencies and in each population [23]. In a study by Sha-banowitz et al., most employees (60%) believed that giving up their jobs during a pandemic is unethical because of their duty to care, while 65% wanted autonomy in deciding whether to work or not, although 79% would agree to volunteer with some incentives such as protective equipment and training in communicable diseases [24]. In the same study, 64% of respondents declared their readiness to report on-call duty in the event of a disaster, even if the respondents were not asked to do so. Most often they were respondents aged 55 and more (74%). The respondents whose length of service was in the range of 16-20 years (74%) also agreed more often than the rest of the respondents to take on duty in the event of a disaster [21, 24].”
“…The current COVID-19 pandemics enforced a rotation between diverse specialties to cover the need for physicians [35]. It also highlighted the moral and ethical issues of COVID-19 infected patients by physicians who lack the necessary knowledge and prac-tical skills As Dawid Orentlicher notes, not every physician can be an emergency room physician dealing with emergencies [36]. The lack of appropriate training is a significant challenge, especially for younger physicians, as indicated by Ayub et al. [34]. Therefore, physicians without specialist training and those who are uncertain about their competency should not be forced to and delegated duties related to intensive care of a patient. Ad-ditionally, the World Health Organization (WHO) interim guidelines for COVID-19 allow health care workers to exercise their right to remove themselves from their work situation if they have reasonable grounds to do so [37]. The guideline is based on the opinions of healthcare workers on the readiness to be transferred to other departments in the event of a catastrophe. In this study, 37% of the respondents agreed, and 39% did not agree to be transferred. Most often, respondents aged 35-44 (43%) and respondents with 11-15 years of experience (50%) did not consent to such a transfer.
In summary, although healthcare professionals have an obligation to get involved in the treatment of patients in hazardous environments and severe conditions such as those with COVID-19, the capabilities and limitations of such involvement should be critically assessed, making such involvement conditional. Research shows how important it is to respect the rights and interests of all parties involved in a hazardous situation such as a pandemic. The core principle in working disasters and emergencies is the MIMMS 3S in safety; Self Safety, Scene Safety, and Survivors Safety [38]….”
Thank you again for your helpful comments and for taking the time to point out options to improve our manuscript.
Reviewer 2 Report
Dear Editor, while thanking you for the opportunity to judge the article, the following comments suggest to improve the quality of the article:
1- The introduction is very incomplete and does not have a good coherence. The necessity of study is not mentioned
2- In the working method section, the sampling method, how to calculate the sample size?, and the formula for calculating the sample size should be mentioned. Inclusion and exclusion criteria should be mentioned
3- In the method section, no explanation has been given on how to check the validity and reliability of the questionnaires used?. It is necessary to provide more complete explanations about the questionnaires used. What items did each of the Moral, Ethical, Personal, and Professional Challenges questionnaires include?
- How are the relationships between the variables Moral, Ethical, Personal, and Professional Challenges examined? Why in the results section about the relationship between the mentioned variables is not reported?.
5- If your method of work is descriptive study, you should not use the word experiences in the title because the study of experiences is related to qualitative studies.
6- The discussion is very poorly written and does not have good coherence. It is necessary to expand the discussion
- What are the strengths and limitations of reading?
What is the application of the findings in the practice?
Author Response
Response to Reviewer #2
Manuscript ID: ijerph-1699808
Title: The Moral, Ethical, Personal, and Professional Challenges Experienced by Physicians during the COVID-19 Pandemic
Authors: Krzysztof Goniewicz, Mariusz Goniewicz, Anna Włoszczak-Szubzda, Dorota Lasota Frederick M. Burkle, Marta Borowska-Stefańska, Szymon Wiśniewski and Amir Khorram-Manesh
May 2015
Dear Editor,
We would like to express our sincere thanks to you for the positive and constructive comments on our previous manuscript. Based on your comments, we have worked to improve the paper. Our response is provided below, with reviewer comments in italics followed by our response in plain text.
Best regards,
Authors
Comment 1 1- The introduction is very incomplete and does not have a good coherence. The necessity of study is not mentioned
Response: Many thanks for your thoughts. We agree with this suggestion. In our revised paper, we have added information that, in our opinion, strengthens the introduction section.
At the beginning we added “While immediately perilous to the health and lives of infected patients the SARS-COV-2 (COVID -19) pandemic also raised both crucial and yet unknown dangers that could impede the safety of caretakers leaving them feeling uncertain, unsafe, and vulnerable. These resulted in a myriad of critical discussions among providers regarding moral and professional obligations for care [1],”
And the introduction was also extended by a paragraph referring to
“…The current COVID-19 pandemic may have impacted the working situation for all healthcare professionals in various ways. In Ireland, it was a blessing since it improved the working condition as described by interviewees in a study that recognized more doctors staffing the hospital wards during the first wave of the pandemic, and other positive implications for a range of factors crucial for their experience of work, such as the ability to take sick leave, workplace relationships, collective workplace morale, access to senior clinical support and the speed of clinical decision-making [12]. These organizational improvements have been reported by other researchers, who also mentioned several changes in physicians ’psychosocial work environment due to increased workload, and information overload, as well as ethical considerations, uncertainty, making working environment stressful for physicians [13]. Additionally, a recent publication, investi-gating the ethical viewpoints of both civilian and military populations facing diverse scenarios, including pandemics, showed differences between both populations but also within each group. One significant factor influencing ethical viewpoints was participants' nationality (Polish vs. Swedish), but also gender, and professions [14]…’
Comment 2: In the working method section, the sampling method, how to calculate the sample size?, and the formula for calculating the sample size should be mentioned. Inclusion and exclusion criteria should be mentioned. In the method section, no explanation has been given on how to check the validity and reliability of the questionnaires used?. It is necessary to provide more complete explanations about the questionnaires used. What items did each of the Moral, Ethical, Personal, and Professional Challenges questionnaires include? How are the relationships between the variables Moral, Ethical, Personal, and Professional Challenges examined? Why in the results section about the relationship between the mentioned variables is not reported?. If your method of work is descriptive study, you should not use the word experiences in the title because the study of experiences is related to qualitative studies.
Response: Thanks for pointing this out. We agree to these comments. First the methodology was expanded, and we explained missing details. We have added the keys just like the excluded and included criteria. As for the size of the sample, it is something that is hard to explain. The questionnaire was sent by the internal system of the medical university in which the doctors have registered accounts. We are not able to determine how many such people are there due to the protection of personal data. It is also difficult to diagnose how many people actually received such a message - i.e. how many opened it or how many doctors simply ignored it.
We add this information into the text:
The questionnaire was developed based on a literature review by all authors. The following search engines were used to search for literature: PubMed, Scopus, and Web of Science. Using the following keywords: “COVID-19” AND “ethics” AND “professional attitude” AND “physician”, a high number of hits was obtained. The search was then limited to the publications in English, then researchers conducted the search inde-pendently, and the outcomes were then matched.
Included studies: Original publications and reviews from January 2010 to 2022
Excluded studies: Proceedings, editorials, meeting notes, news, abstracts, and non-relevant papers.
Finally, a qualitative thematic analysis of the included literature based on an in-ductive approach was applied. This content analysis aimed to study all included articles, focusing on similarities and differences in the findings to build the questionary.
The acquired data were then organized, categorized, and mapped. The question-naire-based on the descriptive research method consisted of 9 questions and was con-structed to be completed in 5 minutes. There were four questions, which aimed to assess the perceived preparedness quantitively. These questions aimed to quantify perceived ethical issues. Each question in this group was formulated as a statement, which could be answered using a Likert scale from 1 to 5, where 5 meant ‘agree’, 4 ‘disagree’ and 5 ‘no opinion’. The remaining questions were demographic. To verify the research tool, the questionnaire was tested on a sample of 15 employees in one university hospital. This group was then excluded from the study and their answers were not included in the final analysis. The outcome was reviewed based on a combination of logic, relevance, com-prehension, legibility, clarity, and usability. In the initial phase of the tool verification, the questionnaire was tested on 20 physicians of the Medical University of Lublin who were then excluded from the main study.
Comment 3: The discussion is very poorly written and does not have good coherence. It is necessary to expand the discussion
Response: Thank you very much for your suggestions. That was very important and we believe we have greatly strengthened the discussions with new paragraphs as well as modifications to the previous ones. The following parts has been added:
“…This study confirms the acceptance of personal sacrifice among Polish physicians during their professional duties. However, not everyone in this group was prepared to work unconditionally and accepted the personal risk of infection during a pandemic. Similar results have been published for other healthcare professionals and necessitate new considerations regarding the professional working environment before an increasing number of future public health emergencies [15].Pursuing a medical profession is associated with social trust as well as a duty and/or sacrifice. Therefore, it is commonly believed that the ideological foundation of the medical profession lies in the “altruism”, which is defined as giving priority to the benefits of others, with the possibility of one's losses [16-17]. This belief is also recognized in the Hippocratic Oath, which is the basis of today's medical ethics, requiring a physician to commit: "I will use ... my greatest abilities and judgment, and I will not do any harm or injustice" [18]. The American College of Physicians Handbook of Ethics states further that the ethical imperative of providing physician care overrides the risks to the physician even in a pandemic. The American Society of Infectious Diseases, as well as the American Medical Association (AMA), also emphasize physicians’ obligation to treat, even at the risk of contracting the patient's disease, or when faced with greater than usual risks to their safety, health, or life [19-20]. Similar guidelines can be found in Good Medical Practice in Great Britain, where the General Medical Council obliges the physician to treat cases of high risk [21].Although there are several guidelines and instructions, the perception of those facing the real threat at the operational level should also be considered. In a study by Sultan et al. [22], there were clear differences in healthcare staff’s willingness to work in different disaster and public health emergency scenarios. Additionally, Khorram-Manesh et al. have shown that ethical and moral viewpoints of both civilian and military healthcare staff facing various scenarios are not predictable, emphasizing a need for both synchronizing these views between diverse agencies and in each population [23]. In a study by Sha-banowitz et al., most employees (60%) believed that giving up their jobs during a pandemic is unethical because of their duty to care, while 65% wanted autonomy in deciding whether to work or not, although 79% would agree to volunteer with some incentives such as protective equipment and training in communicable diseases [24]. In the same study, 64% of respondents declared their readiness to report on-call duty in the event of a disaster, even if the respondents were not asked to do so. Most often they were respondents aged 55 and more (74%). The respondents whose length of service was in the range of 16-20 years (74%) also agreed more often than the rest of the respondents to take on duty in the event of a disaster [21, 24].”
“…The current COVID-19 pandemics enforced a rotation between diverse specialties to cover the need for physicians [35]. It also highlighted the moral and ethical issues of COVID-19 infected patients by physicians who lack the necessary knowledge and prac-tical skills As Dawid Orentlicher notes, not every physician can be an emergency room physician dealing with emergencies [36]. The lack of appropriate training is a significant challenge, especially for younger physicians, as indicated by Ayub et al. [34]. Therefore, physicians without specialist training and those who are uncertain about their competency should not be forced to and delegated duties related to intensive care of a patient. Ad-ditionally, the World Health Organization (WHO) interim guidelines for COVID-19 allow health care workers to exercise their right to remove themselves from their work situation if they have reasonable grounds to do so [37]. The guideline is based on the opinions of healthcare workers on the readiness to be transferred to other departments in the event of a catastrophe. In this study, 37% of the respondents agreed, and 39% did not agree to be transferred. Most often, respondents aged 35-44 (43%) and respondents with 11-15 years of experience (50%) did not consent to such a transfer.
In summary, although healthcare professionals have an obligation to get involved in the treatment of patients in hazardous environments and severe conditions such as those with COVID-19, the capabilities and limitations of such involvement should be critically assessed, making such involvement conditional. Research shows how important it is to respect the rights and interests of all parties involved in a hazardous situation such as a pandemic. The core principle in working disasters and emergencies is the MIMMS 3S in safety; Self Safety, Scene Safety, and Survivors Safety [38]….”
Comment 4: What are the strengths and limitations of reading?
Response: Thank you very much for your comment. The Limitation section has been added to the paper with following information “The main limitation of this study was the limited number of physicians from one city (the city of Lublin). The COVID-19 pandemic proved to be an obstacle to further research, which could have impacted response rates and might have generated response bias. Another limitation was the lack of consideration for illness, personal or chronic condition, pregnancy, pressure from family or spouse, or other factors that might have influenced the responses. Additionally, there was no space for free text in the questionnaire, thus respondents did not have the opportunity to add topics they felt were of relevance. Despite these limitations, this study opens a discussion on this subject and the need for broader research in this area. Due to the essence of the problem, and its related consequences, further in-depth research is recommended. At the same time, this study serves as wider standardization of the research tool used.
“
Thank you again for your helpful comments and for taking the time to point out options to improve our manuscript.
Round 2
Reviewer 1 Report
Dear authors,
Thank you for the improvements made to the text. However, it seems to me that some clarifications could be made:
INTRODUCTION
- The authors did not respond to suggestions for clarification of the study objective: The authors should specify in the 2nd objectives they will evaluate whether there are differences in attitudes of physician based on age and seniority
MATERIALS AND METHODS
- The authors should clarify that this is a cross-sectional survey
- The authors should specify which demographic variables are collected
- Details of the software used for the online questionnaire are missing.
- The authors have indicated that the number of physicians eligible for the study should not be specified. The authors should clarify that this is a weakness of the study in determining the participation rate and assessing the representativeness of the sample (discussion)
- The authors could have specified that they would investigate whether there was a significant difference attitudes of physicians by age group and by seniority.
RESULTS
- Tables 2 and 3: It is difficult to identify significant results by 5 categories from the presentation of the results in Tables 2 and 3. Why not present the results in the tables in 3 categories as in the text(I have no opinion; I disagree/rather disagree; I agree/rather agree)?
- The presentation of the results in Tables 2 and 3 does not directly correspond to the objective formulated in the introduction but rather to a distribution according to age groups and seniority. The authors are advised to clarify the objective
DISCUSSION:
- The authors could enrich the discussion by presenting the differences observed in the study according to age and seniority.
- The authors could discuss the weaknesses of the study in relation to the choice of a cross-sectional survey, the representativeness of the sample, the declarative mode of information collection.
Author Response
Response to Reviewer #1
Manuscript ID: ijerph-1699808
Title: The Moral, Ethical, Personal, and Professional Challenges Experienced by Physicians during the COVID-19 Pandemic
Authors: Krzysztof Goniewicz , Mariusz Goniewicz , Anna Włoszczak-Szubzda , Dorota Lasota , Frederick M Burkle , Marta Borowska-Stefańska , Szymon Wiśniewski , Amir Khorram-Manesh
May 2022
Dear Editor,
We would like to express our sincere thanks to you for the positive and constructive comments on our previous manuscript. Based on your comments, we have worked to improve the paper. Our response is provided below, with reviewer comments in italics followed by our response in plain text.
Best regards
Authors
Comment 1: The authors did not respond to suggestions for clarification of the study objective: The authors should specify in the 2nd objectives they will evaluate whether there are differences in attitudes of physician based on age and seniority
Response: Thank you for your comment. We apologise that we did not include this comment previously. We have add to our objective that we conduct this research based on age and seniority.
Comment 2: The authors should clarify that this is a cross-sectional survey
Response: Thank you for your comments. This information has been added. Please see line 114 for our edits.
Comment 3: The authors should specify which demographic variables are collected
Response: Thank you for your comment. The remaining questions were demographic, and the following variables were collected: age, gender, workplace, and length of service. This information was included in 2.2 Questionnaire subsection
Comment 4: Details of the software used for the online questionnaire are missing
Response: Thank you for your comment. In subsection 2.3 we have add information about software.
Comment 5: The authors have indicated that the number of physicians eligible for the study should not be specified. The authors should clarify that this is a weakness of the study in determining the participation rate and assessing the representativeness of the sample (discussion)
Response: Thank you for pointing this out. We agree and this information was included in limitation section. “Another weakness of the study was the determining the participation rate and assessing the representativeness of the sample as the number of physicians eligible for the study has not been specified.”
Comment 6: The authors could have specified that they would investigate whether there was a significant difference attitudes of physicians by age group and by seniorit
Response: Thank you for the suggestions. This information has been added to introduction section as a objective.
Comment 7: Tables 2 and 3: It is difficult to identify significant results by 5 categories from the presentation of the results in Tables 2 and 3. Why not present the results in the tables in 3 categories as in the text(I have no opinion; I disagree/rather disagree; I agree/rather agree)?
Response: Thanks. We still believe the present table are presented well and we have consulted it with statistician who conformed they are clear. If you think the text should be changed into similar categories we can do that but the presented clarification in our opinion is
understandable.
Comment 8: The authors could discuss the weaknesses of the study in relation to the choice of a cross-sectional survey, the representativeness of the sample, the declarative mode of information collection.
Response: Thanks for the comment. As mentioned above we have add this information to limitation section
We hope that the manuscript will meet your expectations and that our changes will be satisfactory.
Thank you again for your helpful comments and for taking the time to point out options to improve our manuscript.
Reviewer 2 Report
Dear Editor The author has made complete corrections and the article has been improved. Therefore, the article is worth publishing.
Author Response
Dear Reviewer,
Thank you again for your helpful comments and for taking the time to point out options to improve our manuscript.